# Ultrasensitive Nanophotonic Random Spectrometer with Microfluidic Channels as a Sensor for Biological Applications

**DOI:** 10.3390/nano13010081

**Published:** 2022-12-24

**Authors:** Aleksei Kuzin, Ilia Fradkin, Vasiliy Chernyshev, Vadim Kovalyuk, Pavel An, Alexander Golikov, Irina Florya, Nikolay Gippius, Dmitry Gorin, Gregory Goltsman

**Affiliations:** 1Center for Photonic Science and Engineering, Skolkovo Institute of Science and Technology, 121205 Moscow, Russia; 2 Department of Physics, Moscow State Pedagogical University, 119992 Moscow, Russia; 3Center for Engineering Physics, Skolkovo Institute of Science and Technology, 121205 Moscow, Russia; 4Center for Photonics and 2D Materials, Moscow Institute of Physics and Technology, 141700 Moscow, Russia; 5Laboratory of photonic Gas Sensors, National University of Science and Technology MISiS, 119049 Moscow, Russia; 6Tikhonov Moscow Institute of Electronics and Mathematics, National Research University Higher School of Economics, 101000 Moscow, Russia; 7Quantum Photonic Integrated Circuits Group, Russian Quantum Center, 143025 Skolkovo, Russia

**Keywords:** nanophotonic random spectrometer, photonic integrated circuits, microfluidics, ultrasensitive sensor, biological and theranostic applications

## Abstract

Spectrometers are widely used tools in chemical and biological sensing, material analysis, and light source characterization. However, an important characteristic of traditional spectrometers for biomedical applications is stable operation. It can be achieved due to high fabrication control during the development and stabilization of temperature and polarization of optical radiation during measurements. Temperature and polarization stabilization can be achieved through on-chip technology, and in turn robustness against fabrication imperfections through sensor design. Here, for the first time, we introduce a robust sensor based on a combination of nanophotonic random spectrometer and microfluidics (NRSM) for determining ultra-low concentrations of analyte in a solution. In order to study the sensor, we measure and analyze the spectra of different isopropanol solutions of known refractive indexes. Simple correlation analysis shows that the measured spectra shift with a tiny variation of the ambient liquid optical properties reaches a sensitivity of approximately 61.8 ± 2.3 nm/RIU. Robustness against fabrication imperfections leads to great scalability on a chip and the ability to operate in a huge spectral range from VIS to mid-IR. NRSM optical sensors are very promising for fast and efficient functionalization in the field of selective capture fluorescence-free oncological disease for liquid/gas biopsy in on-chip theranostics applications.

## 1. Introduction

Recently, there has been a steady increase in the trend towards point-of-care (POC) diagnostics in the medical sector of monitoring and treatment [1]. In this regard, there is rising demand for compact sensors that are highly sensitive, reusable, do not require a large amount of analyte, and are capable of monitoring specific analytes in real time.

According to the change of the physical parameters after interacting with molecules in the environment, liquid/humidity/gas sensors can be categorized into many types [2]. There are the capacitance, resistance, impedance, optic-fiber, quartz crystal microbalance (QCM), surface acoustic wave (SAW), resonance type sensors, and so on [3]. These sensors are based on different platforms such as ceramics, semiconductors [4], perovskite compounds [5], polymers [6], carbon nanotubes [7], graphene [2,8], and others. Optical affinity biosensors based on resonant dielectric nanostructures are the most perspective among biosensors due to their peculiarity of using an evanescent-field exponential decay along an axis perpendicular to the sensing surface, with a decay length on the order of hundreds of nanometres for scanning analytes [9]. This surface confinement by the microfluidic channel is an important feature that increases the light–matter interaction and offers precise spatial control over the measurement. The combination of photonic integrated circuits (PICs) with microfluidic channels is one of the promising directions among affinity biosensors. The advantages of this combination are, on one hand, provided by planar and CMOS compatible fabrication technology, high sensitivity, and low limit of detection (LOD), and on the other hand, by the possibility of pumping analytes and modifying the surface of sensitive areas of PICs for increasing specificity to determined analytes by using microfluidic channels. A lot of work related to these advantages has already been presented [10,11], among which devices can be distinguished by the chosen sensitive component, the microring resonators (MRRs), and Mach–Zehnder interferometers (MZIs). The operation principle of these devices is based on detection by the PICs due to a change in the refractive index of the medium above the sensitive area of the sensor. However, in the case of MRRs and MZIs, despite the high values of sensitivity and low LOD, there is no possibility of reconfiguring the system depending on defects and residues for unwashed analytes and directly detecting the analyte spectrum. That is why it is necessary to use PIC spectrometers with reconfigurable spectrum characteristics as detectors.

A spectrometer is a basic and widely used optical instrument for measuring the different spectrums of the entering optical radiation. Traditional spectrometers are usually based on interferometers of various types such as diffraction grating or Fabry–Perot interferometry. The basic principle of their operation is related to multipath interference when optical radiation propagates in different directions depending on the wavelength. Thus, the relative location of the interferometer and the detector uniquely determines which wavelength of optical radiation measured it. The most widespread spectrometers among PIC are based on an array of planar waveguides of various lengths (arrayed waveguide gratings: AWGs) [12]. Such spectrometers in combination with microfluidic channels are very promising for fluorescence spectroscopic analyses and Raman spectroscopy, which can be very useful in the POC approach. However, these spectrometers cover a relatively large area on a chip surface and are extremely sensitive to fabrication deviations. For example, a small deviation in the waveguide width leads to a change in the waveguide mode index and, accordingly, a violation of the interference conditions, which makes such devices difficult to reproduce. For this reason, the so-called random spectrometers are extremely favorable for on-chip implementation [13]. Their main difference is that the interference in this device is random and occurs due to scattering by various disordered elements and dots. Thus, the probability of optical radiation of a certain wavelength hitting the detector becomes a random variable depending on the position of the scatterers and their configuration. This leads to the fact that previously, the presence of a signal on a certain detector unambiguously signified the presence of a corresponding spectral line, but now a special, non-trivial mathematical processing of seemingly random data is required to obtain the spectrum.

In this work, for the first time, we report on a robust sensor based on the combination of the nanophotonic random spectrometer and microfluidic technology for determining ultra-low concentrations of isopropanol in DI water. In order to study the sensor, we measure and analyze the spectra of different isopropanol solutions of known refractive indexes. Simple correlation analysis shows that the measured spectra shift with a tiny variation of the ambient liquid optical properties reaches a sensitivity (*S*) of approximately 61.8 ± 2.3 nm/RIU. These optical sensors show promise for biomedical applications due to their robustness against fabrication imperfections, ability to be used in different spectral ranges, and easy scalability on a chip.

## 2. Materials and Methods

NRSM fabrication can be divided into the following main steps: development and fabrication of a photonic chip with random spectrometers and its combination with microfluidic channels. That is why the final configuration of NRSM included a nanophotonic chip, polydimethylsiloxane (PDMS) with formed MFCs (microfluidic channels), copper support of the chip, plastic holder, and fluid tubes (see Figure 1a).

The spectrometers were fabricated on commercially available silicon nitride wafers with a 450 nm silicon nitride layer on top of a 2.6 μm SiO_2_ buffer, formed on the silicon (Si) substrate. The random structure, photonic-crystal boundary, and the focusing grating couplers (FGCs) [14] were all defined during a single electron-beam lithography exposure. After that reactive and ion etching (RIE) was performed for pattern formation. The scattering media consisted of randomly situated 200 nm radius air cylinders (see Figure 1c). A half-etched design for forming waveguide structures was used for supporting TE polarized light in the wavelength range of operation from 1510 to 1620 nm (see Figure 1b). FGCs also supported TE polarized light. TE polarization scatters more in the forward direction due to scattering by *p* waves being dominant. Therefore, the light is primarily detected in the central output ports as long as the diffusive regime is not fully developed [15]. In the case of TM polarization, scattering by *s* waves will be dominant. It leads to more isotropic single-particle scattering, and subsequently, the light is more evenly distributed over all output ports even if the diffusive regime is not fully developed.

The experimental setup consists of three main parts (see Figure 1d): optical parts, microfluidic parts, and data cables. A microscope with a built-in camera is installed above the array of fibers, which is necessary to align the adjustable stage with the array of fibers. The polarization controller (FPC) consists of three rotating plates that control the polarization of the incident light. By changing the rotation angle of the plates, the position of the controller is set to where the maximum optical transmission on the detector is achieved. The following method was used to test the NRSM: initially, the photonic chip (sample) was installed on an adjustable stage with piezo motors. After that, the optical radiation from the laser was propagated through the optical fiber toward the PIC. Then, the polarization controller was directed through the fiber array to the input FGC on the chip. Optical radiation was passed through the PICs and the FGC was outputted to the fiber array. The graph of the output spectrum was generated by using the LabVIEW program and displayed on a personal computer screen. Only the optical fibers (orange color) and data cables (black color) are necessary for the nanophotonic device calibration test. To test different analytes, it was necessary to package the chip with a microfluidic system (blue color). The microfluidic system consisted of a peristaltic pump, hydrophobic linking tubes, and PDMS with formed MFCs.

## 3. Results

### 3.1. Developing RDMX Design

Achievement of high resolution in any spectrometer requires a very large optical path to accumulate a significant phase shift between close spectral lines. In terms of integrated devices, this concept is not very convenient due to increasing the device’s size on a chip, which requires precision during the fabrication on such scales. One of the most poignant ways to overcome this obstacle is to force optical radiation to travel a long distance on a small footprint by reflecting from densely located scatters (holes in a waveguide). The large number of randomly distributed holes half-etched in the waveguiding layer scatter the photonic modes in different directions. As a result, optical radiation travels for a long time until it escapes from the structure. This allows to us achieve the long optical paths of interfering “beams” and the narrow width of typical peaks in spectra via a structure of a relatively small footprint. Due to the combination with a microfluidic channel, it becomes possible to limit the space above the sensitive part of the spectrometer and to pump various liquids and gases atop. A change in the refractive index of the medium due to different solutions in the channel will lead to a varying effective refractive index, which in turn will correct the optical path and the speckle pattern. The design of our sensor is inspired by the concept of the integrated random spectrometer [15,16].

Initially, the literature was reviewed and the considered structure was analyzed theoretically. First of all, it turned out that in a diffusive regime the difference in the optical path between the first and the last beam is of the same order as the mean optical path of beams that interfere. Knowing that it is the process of random walks, the root mean square distance from the start is equal to: (1)$$R=\sqrt[{}]{N}l,$$
where *N* is the number of scatterings and *l* is the mean free path. Assuming that *R* is the radius of the structure, the typical optical path gained in the diffusive regime before escaping from the structure is the following: (2)$$L_{\mathrm{dif}}=Nl=\frac{R^{2}}{l^{2}}l=R^{2}/l.$$

In the diffusive regime, the optical path grows quadratically with a linear size of a device, and randomly distributed scatters are not sensitive to fabrication defects. The resolution of an interferometer is proportional to the difference in optical paths between the interfering beams over the wavelength of light RES≈ΔL/λ [17]. In our case, this quantity might be estimated as a difference between the diffusive and ballistic paths $\Delta L\approx L_{\mathrm{dif}}-L_{\mathrm{bal}}\approx R^{2}/l-R\approx R^{2}/l$ since R≫l. Finally, we obtain: (3)$$\mathrm{RES}\approx \Delta L/\lambda =\frac{R^{2}}{l\lambda }.$$

For the next step, six spectrometers of different hole diameters *d* (150 nm and 200 nm) and different radii *R* and hole quantity *N* (100 μm (5000), 150 μm (11,450), 200 μm (20,000)) were made (see Figure 2a,b). Twenty detectors potentially allow operating in a spectral range up to 20Δλ, where Δλ is the width of an autocorrelation function of a signal of a single detector. In the experiment, the diffusive regime was not fully reached and the central detectors worked in ballistic mode (see Figure 2c). However, it is possible to estimate the working performance by calculating the correlation matrix. Each element of the matrix is easily calculated by the relation $r_{ij}=\frac{\int I_{i}\left(\lambda \right)I_{j}\left(\lambda \right)d\lambda }{\sqrt{\int I_{i}^{2}d\lambda \int I_{j}^{2}d\lambda }}$. Ideally, different detectors should not correlate with each other to provide independent information about the spectrum of interest (see Appendix A). This condition is almost fulfilled in the experiment for the 90 nm spectral range (see Figure 2d).

### 3.2. Combination RDMX with MFCs

For this stage, the spectrometer with optimal values for *d* = 150 nm, *R* = 150 μm, and *N* = 11,450 were chosen. From the spectrum of one of the output channels (see Figure 3) for two liquids of slightly different optical properties, there are multiple resonances and several-nanometer-wide peculiarities that are associated with the random interference mentioned above. As shown in Figure 3, “high-frequency” oscillations (not resolved in the graphs) are most probably associated with Fabry–Perot oscillations between the surface of a chip and fiber facet.

The considered structure supports 11 output channels providing unique sharp spectra. As a result, even a small deviation in the refractive index of the studied liquid significantly affects the optical spectra of the structure in corresponding spectral ranges. The list of such approaches includes many advanced computational methods that can be applied to restore the change of the refractive index from the obtained spectra. Examples include compressed sensing techniques, singular value decomposition-based methods, genetic algorithms, and others. Nevertheless, in this study, we only demonstrate the operational feasibility via the simple correlation analysis.

Indeed, we can assume that the small change in the liquid refractive index implies a linear shift of all the spectra of the structure. In order to determine the relative shift of two spectra we calculate the correlation function between them. Obviously, the relative shift can be found from the position of the maximum of the correlation function.

Nevertheless, as we can see in Figure 4, which shows several correlation functions, the high-frequency oscillations seem not to be shifted at all. Therefore, the global maxima of some correlation functions can be pinned to zero. This means that the physical process underlying the “frequent” interference is not affected by the liquid and is likely to be a coupling-assisted effect. In order to get rid of this peculiarity and determine the “true” position of the maximum, we just smooth the autocorrelation functions with Gaussian filters. The width of the filter should be larger than a period of Fabry–Perot oscillations, but not too large to blur the random-scattering-associated phenomena. In our case, the optimal standard deviation (width) of the filter is approximately σ=0.6 nm. Such a filter width was used in all the presented calculations.

Finally, we compute the correlation functions for all the combinations of measurements for 17 different concentrations of isopropanol (ranging from 80 ppm to 100,000 ppm), smooth them, and find the positions of maxima, which correspond to the relative shift of corresponding spectra. Since the spectral shift is small enough, we can assume that it depends linearly on the refractive index of the considered solution. In turn, this allows us to plot the relative shift as a function of the difference between the corresponding refractive indices (Figure 5). For each point in the graph, there is another one that corresponds to the opposite refractive index difference and wavelength shift. This is a representation of an obvious fact that if one of the spectra is red-shifted to the second one, then the second spectrum is blue-shifted to the first.

## 4. Discussion

The spectral resolution of the random spectrometer depends on the change in wavelength that is required to generate an uncorrelated intensity distribution on the detectors [16]. As we can see from Figure 5, there is a clear linear dependence of the spectra shift on the difference between the refractive indices of the considered liquids. The slope angle (*S*) of this linear curve was found to be −61.8 ± 2.3 nm/RIU. The dependence of the sensitivity on a temperature variation is discussed in the Appendix A. This proves the possibility to utilize the considered device for the detection of the refractive index change, but the potential error is rather large. Obviously, the precision might be significantly improved via the application of more accurate data processing techniques and mainly via accounting for the spectra from all the output channels.

Nevertheless, due to the operation peculiarity, the random spectrometers have a number of significant advantages. First, such devices are not sensitive to fabrication deviations. Since their operation is based on random scattering, a random change in the structure will naturally be taken into account in the calibration and will not require any additional processing. Secondly, random spectrometers can be used in different spectral ranges simply by appropriate data processing. Thirdly, the random spectrometer is very easy to scale: by increasing the size and number of detectors, the resolution and the working spectral range can be consistently increased.

Another advantage of the random spectrometer is that it can operate over an extremely broad frequency range without structural modification. In current work a Gaussian profile of input optical radiation which was coupled in focusing grating coupler for C-band range [14] on a chip was used. That is why the current on-chip spectrometer works only for a fixed spectral range. In case of butt coupling the random spectrometer works in the visible and NIR wavelength range due to the use of the Silicon Nitride layer as a waveguide. The scattered light for any beam profile and irradiance will always reach the detector array, regardless of its frequency. The switch of operation frequency is done simply by changing the transfer matrix to the one calibrated for the target spectral range [13]. One more interesting realization of a sensor based on the speckle pattern processing approach was presented in [18]. Authors demonstrated a highly sensitive technique in free space for evaluating surface tension and dynamic viscosity of nanofluids. This approach can be used in combination with what is presented in the current work. NRSM not only can be used for determining surface tension and dynamic viscosity properties by mechano-optical measurements, but also for determining optical properties of analytes.

Additionally, with the appropriate parameters of the device, it is possible to change its resolution simply by changing the width of the working spectral range.

## 5. Conclusions

In this paper, for the first time, the results of a combination of a random on-chip spectrometer and microfluidic channels were presented. This device has been characterized in terms of sensitivity by using different concentrations of isopropanol in DI water. Based on the data obtained, this sensor after refinement can be promising in terms of controlling surface functionalization and then specific detection of analytes due to its universal reconfigurability.

## Figures and Tables

**Figure 1 nanomaterials-13-00081-f001:**
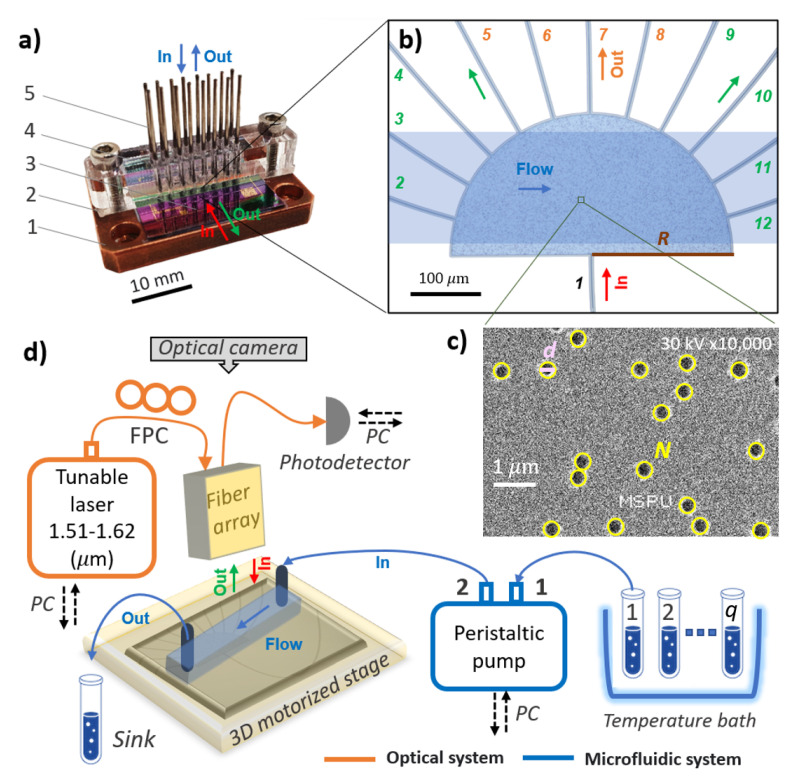
(**a**) Image of the microfluidic nanophotonic sensor: 1—copper support of the chip, 2—PDMS with formed MFCs, 3—nanophotonic chip, 4—plastic holder, and 5—fluid tubes. (**b**) Optical image of random spectrometer without background. The orange color for output channels represents the ballistic regime of propagation for optical radiation, and the green color represents the diffusive regime; *R* is the random spectrometer radius. (**c**) The image of holes obtained by using an electron microscope; *d* is the hole diameter; *N* is the hole quantity. (**d**) Schematic view of the experimental setup: PC—personal computer, FPC—fiber polarization controller, 1 2 … *q*—different concentrations of isopropanol in DI water.

**Figure 2 nanomaterials-13-00081-f002:**
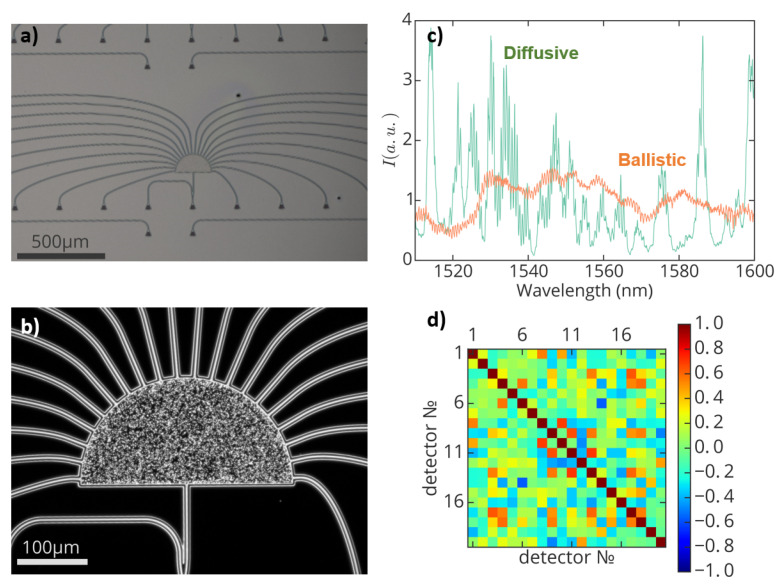
(**a**) Bright-field image of the integrated random spectrometer (**b**) Dark-field image of integrated random spectrometer (**c**) Response signal on detectors operating in the diffusive and ballistic regimes. The orange color for the transmission spectrum represents the ballistic regime of propagation for optical radiation and the green color represents the diffusive regime. (**d**) Correlation matrix for a spectrometer, which does not account for the poorly calibrated region.

**Figure 3 nanomaterials-13-00081-f003:**
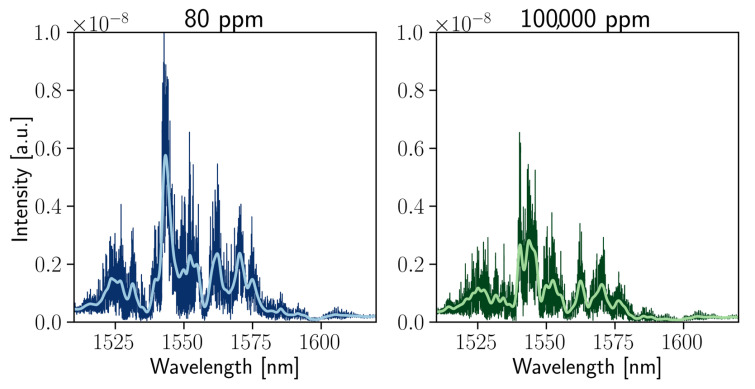
Spectra of the considered structure for two different concentrations of isopropanol solutions in water. Dark lines correspond to the originally measured spectra, whereas light lines correspond to the smoothed data via a 0.6 nm width Gaussian filter.

**Figure 4 nanomaterials-13-00081-f004:**
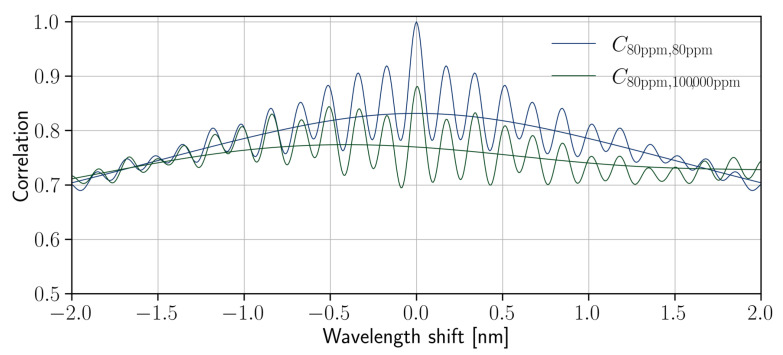
Correlation functions between spectra of different solutions and after their smoothing with a 0.6 nm Gaussian filter. Blue lines correspond to the autocorrelation function of the 80 ppm solution (almost pure water). Green lines correspond to the correlation of the 80 ppm and 100,000 ppm solutions’ spectra. It can be observed that the peak of the smoothed green line is significantly shifted from the origin.

**Figure 5 nanomaterials-13-00081-f005:**
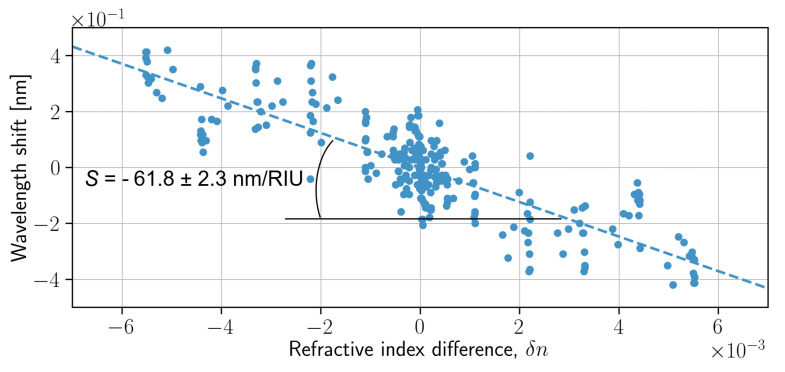
Dependence of the spectra shift on the refractive index deviation.

## Data Availability

Appendix A include the data about optimization of a random spectrometer design and dependence of sensitivity on temperature variation.

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
