# Peer review of "Ultrasensitive Nanophotonic Random Spectrometer with Microfluidic Channels as a Sensor for Biological Applications"

_nanomaterials, 2022, doi:10.3390/nano13010081_

Round 1

Reviewer 1 Report

In this manuscript the authors introduce a robust sensor based on a combination of nanophotonic random spectrometer and microfluidics (NRSM) for determining ultra-low concentration of analyte in solution. In order to study the sensor, author measure and analyze the spectra of different isopropanol solutions of known refractive indexes.

Although this Research work seems interesting, but before publishing, I urge the authors to make the following improvements considering my primary recommendations.

1-      The scale bar of the SEM image presented in Fig 1c needs to be improved and make it readable for the readers.

2-      For the resolution of the structure, the author gave a mathematical equation 3 but did not gave any reference. It will be good if author can give some references about the Equation 1, 2, or 3.

3-      The author claimed that for 17 different concentrations of isopropanol (ranging from 80 ppm to 100000 ppm) the relative  maxima shift of corresponding spectra. What is the reason of this shift? Please give some details or references to prove this claim.

4-      The recently reported articles about biomolecule must be cited in the introduction section on the appropriate places. (Nanomaterials (Basel). 2019 Mar; 9(3): 422.) & ( https://doi.org/10.1002/adfm.202204781), and (https://doi.org/10.1016/j.jallcom.2022.165815).

5-      The minor English language corrections should be made before the revision submission.

Remarks: Publish after Major revision

Author Response

Dear reviewer, thanks a lot for your critical and important comments. We have focused on preparation full answers for all yours proposed points. You can find our answers for your suggestions (comments) and other reviewers in the applied file.

Reviewer 2 Report

Good paper further developing the already described random photonic spectrometer, combined with a microfluidic system. Excellent design and construction, but meaningless evalaution of performance on isopropaol:water mixtures. The main use of such system will be photonic biosensing, but this is not evaluated or tested here. A routine antibody-antige interaction should be tested and results compared with alternatives; is this construction better or worse considering real biosensing use? Nobody is interesting in simulations and low concentrations of isopropanol. The content does not correspond to the chosen journal section.

Author Response

(The authors gave the same response as above.)

Reviewer 3 Report

The manuscript ID nanomaterials-2102233 is an original article that mainly presents a study about a particular sensor based on a combination of nanophotonic random spectrometer and microfluidics for determining ultra-low concentration of some analytes in solution. Please see below a list of comments to the authors:

1. In my opinion, the introduction ought to be enhanced in order to see a more panoramic envision of the topic of research that easily justify the aim of the work.

2. It is not clear how were selected some of the parameters described in the Materials and Methods section. Please describe in order to see that the design is systematic instead of incidental.

3. It should be described the influence of the polarization in the main findings.

4. The dynamic inelastic response exhibited by organic systems in fluids evolves according to different parameters, you can see for instance: https://doi.org/10.1364/OE.26.002033. The authors are invited to discuss about potential perspectives and future research with these considerations.

5. Advantages and disadvantages of the proposed system should be confronted with updated publications in comparative topics. You can see for instance: https://doi.org/10.1038/s41565-021-01045-5

6. Please add the citation to guarantee the proper use of the equation employed to estimate the resolution.

7. Error bar in experiments should be provided.

8. Is the sensitivity of the instrument dependent on temperature?

9. How is the importance of the beam profile and irradiance of the laser system in the main results?

10. Please include other key parameters in figure 1 to see the main considerations analyzed in this work.

Author Response

(The authors gave the same response as above.)

Round 2

Reviewer 1 Report

Accept it in the present form.

Reviewer 3 Report

The authors have provided a successful revision of the manuscript. In my opinion the results are solid and the analysis is clear. The conclusions can be useful for future research and then I can recommend this work for publication in present form.